# Perceptions, barriers, and facilitators of maternal health service utilization in southern Ethiopia: A qualitative exploration of community members' and health care providers' views

Amanuel Yoseph[1]*, Wondwosen Teklesilasie[1], Francisco Guillen-Grima[2,3,4], Ayalew Astatkie[1]

1 School of Public Health, College of Medicine and Health Sciences, Hawassa University, Hawassa, Ethiopia, 2 Department of Health Sciences, Public University of Navarra, Pamplona, Spain, 3 Healthcare Research Institute of Navarra (IdiSNA), Pamplona, Spain, 4 CIBER in Epidemiology and Public Health (CIBERESP), Institute of Health Carlos III, Madrid, Spain

* amanuelyoseph45@gmail.com

**Data Availability Statement:** "All relevant data are within the paper and its supporting information files."

## Abstract

### Introduction

Maternal health service (MHS) use is a key strategy to reduce maternal mortality. However, evidence is scarce in designing efficient intervention strategies in Ethiopia. Thus, we aimed to explore community members and healthcare providers' perceptions of MHS and barriers and facilitators of MHS use in southern Ethiopia.

### Methods

A phenomenological qualitative study was conducted in the month of November, 2022, in the northern zone of the Sidama region. There were sixteen in-depth interviews, nine focus group discussions, and 15 key informant interviews with 112 study participants. A maximum variance sampling method was used to select study participants. Data coding and analysis were done using MAXQDA 2020 software and presented in narratives.

### Results

Communities have positive perceptions and good practices of skilled antenatal care (ANC) and health facility delivery (HFD) but lack awareness of postnatal care (PNC) services and schedules. Some have experienced negative interactions with health care providers, health facilities, and ambulance drivers. The main identified barriers to ANC use were lack of awareness of ANC benefits, distance from a health facility, costs associated with ANC use, long waiting time, lack of road access, and women being busy with different household chores. Distance from health facilities, costs associated with HFD use, unpredicted labor, lack of an ANC visit, lack of a birth preparedness plan, and non-dignified care were the main barriers to HFD. The major barriers to PNC use were home delivery, lack of awareness of

**Funding:** Hawassa University and Sidama president office funded this study. However, funders haven't involved in any stage of research paper.

**Competing interests:** The authors have declared that no competing interests exist.

**Abbreviations:** ANC, Antenatal care; EDD, Expected Date of Delivery; EDHS, Ethiopian Demographic and Health Survey; HCPs, Health Care Providers; FGD, Focus Group Discussion; GTP, Growth and Transformation Plan; H.F, Health Facilities; HEW, Health Extension Worker; HFD, Health Facility Delivery; IDI, In-depth-interviews; KKI, Key Informant Interviews; LMIC, Low and Middle-Income Countries; MHS, Maternal Health Service; MHSU, Maternal Health Service Utilization; MMR, Maternal Mortality Ratio; MPH, Masters of Public Health; PI, Principal Investigator; PNC, Postnatal Care; SSA, Sub-Saharan Africa; WDA, Women Development Army; WDT, Women Development Team; WRA, Women of Reproductive Age.

PNC service and schedule, and socio-cultural beliefs. The main identified facilitators of MHS use were previous experience and fear of obstetric complications, health extension workers and women's development teams, and pregnant women's forums.

## Conclusions

Rural women still encounter challenges when using MHS, even though communities have positive perceptions and good practices of skilled MHS. Bad experiences mothers faced in health facilities, challenges associated with the costs of MHS use, poor awareness of service, and unpredictable labor continued to be fundamental barriers to MHS use. Intervention approaches should consider inter-sectoral collaboration to address community and health facility barriers. The programs must emphasize the transportation arrangements during unpredictable labor and the needs of poor mothers and women with poor awareness of MHS at the community level.

## Introduction

Worldwide maternal mortality is high, and 223 maternal deaths per 100,000 live births (LBs) occurred in 2020 [1]. It will take an annual rate of reduction of 11.6% to bring the worldwide maternal mortality ratio (MMR) below 70 by the year 2030, a rate that has seldom been attained at the country level [1, 2]. Low- and middle-income countries (LMICs) have a disproportionately high MMR (nearly 95% of total maternal mortalities) [3]. Though MMR decreased by almost 34% globally from 2000–2020, significant commitments and efforts are needed in LMICs, particularly in Sub-Saharan Africa (SSA) and Asia, to attain target 1 of Sustainable Development Goal 3 [4, 5].

Ethiopia is one of the countries with a high MMR in SSA [1–3, 5]. According to the report of the 2016 Ethiopian Demographic and Health Survey (EDHS), 412 maternal deaths occurred per 100,000 LBs [6]. Furthermore, great differences in maternal mortality exist across regional states in Ethiopia. For instance, it ranged from 74 in the Tigray regional state to 548 deaths per 100,000 LBs in the Afar region [7]. Similarly, MMR was very high in the southern region of Ethiopia (489/100,000 LBs) [8]. It was also high in the Sidama region (419 per 100,000 LBs), with the Aroresa district having the highest rate at 1142 deaths per 100,000 LBs [9].

Generally, due to different interventions, global maternal survival has increased in the last two decades [4]. Nevertheless, many more survivors suffer from severe situations such as ruptured uterus and obstetric fistula, which can affect them for the rest of their lives [1, 10]. Maternal mortality has remarkable effects on entire families, communities, and nations and is an influence that transcends generations. Complications that cause disabilities and deaths in women also harm neonates and infants they care for [2, 11].

Maternal mortality can be averted if simple preventive measures are considered and sufficient care is accessible and available during critical periods (pregnancy, childbirth, and postpartum) [1, 2]. Besides, maternal health service use (MHSU), comprising access to high-standard quality care, is considered highly effective in decreasing the burden of maternal illness and death, specifically in low-resource areas [1, 2, 4, 12]. Nonetheless, the utilization of the existing MHS is low in developing countries, particularly SSA [2], and there is no exception in Ethiopia [13]. For instance, utilization of at least four ANC visits and proportion of births attended by skilled health care providers ranged between 40 and 60%, while only 20 and 40%

of women had PNC contact with a health professional within two days of delivery in most SSA [14, 15].

For example, the 2019 Mini EDHS report showed that ANC service utilization was 74%; merely 43% of women had four or more ANC visits during their most recent pregnancy; more than half (52%) of all deliveries occurred at home; and only 34% of mothers obtained PNC follow-up within the first 48 hours after delivery in Ethiopia. Besides, significant regional, urban, and rural differences persist in terms of use [13]. Moreover, utilization of MHS was low in the Sidama region, where only 45% of women used at least one ANC, 40.7% attended skilled births, and 14.3% utilized PNC [16].

Several interrelated factors have contributed to low utilization of existing MHS, such as individual, community, socio-economic, and demographic factors; women's obstetric characteristics; organizational or health facility-related factors; health care providers; perceived quality of health services; poor knowledge of obstetric danger signs (ODS); lack of service access; health system functioning; delay in receiving treatment; a dearth of decision-making authority; infrastructure; and socio-cultural and traditional practices [17–26].

The Ethiopian government has been applying multi-dimensional methods, measures, and strategies to halt the low MHSU and universal inaccessibility of maternal health services in line with the principle of primary health care. Some of the strategies comprise the development of a broad 20-year health sector development program [27], a national reproductive health strategy [28], and a growth and transformation plan (GTP) [29], the training and deployment of health extension workers (HEWs) and health care providers (HCPs), particularly midwives in rural areas, the provision of free MHS, free ambulance service to health facilities, expansion of health facility building, and restructured community engagement using the Women Development Army (WDA) [29]. The Ethiopian government has advocated for the free delivery of maternity healthcare services since 2005. The main objective of this advocacy is to make maternity care more accessible to women from various socio-economic backgrounds. Also, by reducing financial barriers that pregnant mothers may encounter, such as the cost of medical cards, medications, laboratory testing, beds, and other expenses, the goal is to improve the utilization of maternity services such as ANC, HFD, and PNC [30].

Despite these efforts being implemented by the Ethiopian government, MHSU was generally low at the country level and very low in rural settings [13]. Hence, the community perceptions and barriers influencing rural mothers to use MHS require a comprehensive understanding in a socio-cultural and socio-economic context. Besides, earlier studies were quantitative [17, 18, 22, 31–34] and recommended conducting further qualitative studies to understand the community's perceptions of MHS and the barriers and facilitators of MHSU. Community perceptions of MHS and barriers and facilitators of MHSU may differ from region to region in Ethiopia, and utilization of MHS is highly variable [13]. Moreover, the existing evidence is limited to the design of effective and efficient, context-specific, locally relevant, and culturally appropriate interventions.

The findings of this study can help inform program managers, policymakers, and implementers regarding where to focus attention in planning intervention strategies to improve MHSU and decrease maternal mortality. Also, the results can be helpful to encourage evidence-based decision-making to address the problems Ethiopian women face throughout the continuum of care. Furthermore, this study can inform maternal health champions by offering community perceptions, barriers, and facilitators working in the context of the Sidama region. Thus, this study explored community members and healthcare providers' perceptions of MHS and barriers and facilitators of maternal health service utilization in the Northern Zone of the Sidama region, Ethiopia.

## Methods

### Study area

The study was done in the Northern Zone of Sidama Region, Ethiopia. Ethiopia is administratively divided into thirteen regions and two city administrations. With a predicted population of about 114 million in 2020, the country will be Africa's second most populous, behind only Nigeria. Sidama region is one of Ethiopia's national, regional states located in the southern part of the country. It was newly established on June 18, 2020, and is the second-smallest regional state, following Harari by land size and the fifth-largest populous in the country [35, 36]. It is located in southern Ethiopia and consists of four zones: the Northern, Southern, Central, and Eastern zones, and one city administration [37]. The northern zone is located 273 km south of Addis Ababa. It has eight districts and two town administrations. The zone has 162 kebeles (Ethiopia's most minor administrative units). Based on a Sidama Region Health Bureau report, the zone has an overall population of 1.29 million. Women of reproductive age (WRA) are estimated to constitute 23.3% of the population. The zone has 144 health posts, 36 health centers, one general hospital, and four primary hospitals. The zone's overall potential health service coverage by public health facilities (HFs) is 70% [38].

### Study design and population

A phenomenological qualitative study was conducted from November 15–30, 2022, among community members and health care providers in the Northern Zone of Sidama Region, Ethiopia. We included all purposefully selected community members and health care providers who lived in the zone for at least six months for the IDIs, KIIs, and FGDs. Midwives who worked for less than two years in the maternal and child health care case team were excluded from KIIs due to their lack of experience, interaction with women, and inability to provide in-depth information. Besides, all WRA involved in the quantitative study conducted parallel with this qualitative study were excluded from IDIs, KIIs, and FGDs because their involvement in the quantitative study may predispose them to information bias. Study participants who had severe illnesses and mental health problems were excluded from IDIs, KIIs, and FGDs because that affected consent procedures and effective participation in this study.

### Basic difference between KIIs and IDIs

The key differences between IDIs and KIIs are based on four parameters: expertise, goal, time, and bias. KIIs are experts. Thus, researchers only conduct KIIs when they can find a person with unique expertise on a topic. IDIs can be conducted with anyone. IDIs are used to learn more about a previously discussed problem. KIIs, on the other hand, are intended to examine a topic before delving into the details. It is often the initial step: figure out what has to be investigated! KIIs often require less time to conduct than traditional IDIs. This is because there are often fewer "experts" to interview than the general population (a sample of participants may be chosen for IDIs). Frequently, KIIs do not accurately represent a population. They are knowledgeable but may not translate well into the actual world. This is something that researchers should keep in mind while selecting KIIs.

### Sample size determination

The sample size for this study was determined based on different recommendations. The sample size for in-depth interviews (IDIs) was determined according to the recommendation of Morse and Creswell for phenomenological studies (5 to 25 study respondents) [39]. Based on their recommendation, we decided to include 21 study respondents. However, we did reach

information saturation after interviewing 13 study participants. We did three further IDIs to ensure the true extent of information saturation. As a result, we finally enrolled 16 study participants. Also, the sample size for focus group discussion (FGD) was determined based on the recommendation (2–3 FGDs with every distinct group of participants with shared features) [40]. We planned to conduct three FGDs per group. We did, but information saturation was not attained until the third FDG per group. Thus, we finally included three FGDs per subgroup: women who gave birth in the 12 months preceding the study, community and religious leaders, and *kebele* leaders. A total of nine FGDs were conducted with 81 participants. This study included nine participants in each FGD from the 6–12 recommended participants [40, 41]. We included medium numbers of FGD participants to avoid the effects of participants who were too small or too large [41]. If the groups are too small, the debate will be slow and frequently directed by one person, and the rich diversity will be lost. Again, if there are too many participants, huge groups are challenging to manage and limit each person's ability to share thoughts and observations.

Furthermore, group dynamics shift when individuals desire but cannot express their experiences. Similarly, the sample size for the key informant interview (KII) was determined based on the recommendation for phenomenological studies (not exceed 35 study respondents) [42]. Based on their recommendation, we decided to include 19 study respondents. We did, however, reach information saturation after interviewing 12 study participants. We did three further KIIs to ensure the true amount of information saturation. As a result, we finally enrolled 15 study participants. We assured information saturation when the same concepts, ideas, and themes were repeated by respondents or no new concepts emerged from the questions.

## Sampling technique

We used a maximum variance sampling method to select eligible community members and health care providers. It is also known as heterogeneous sampling and is one type of purposive sampling method. It is used to capture a broader viewpoint or perspective from different groups of participants as much as possible or to search for variation in viewpoint [43]. Researchers use subjects, cases, organizations, and events that are assumed to be different from those more typical to assure maximum variance. Researchers can find significant common patterns across variants by looking at a subject from various perspectives [43, 44]. In this study, according to the routine administrative data from the district health office, the *kebeles* were classified into better- and lower-performing strata regarding MHSU to ensure variability. Besides, to maintain variability in participants, variability in socio-economic status, age, sex, education status, roles, and responsibility were considered during the selection procedure. IDI and FGD study respondents were recruited from all *kebeles* using pre-defined selection criteria by HEWs, whereas KII participants were chosen with the support of the district health office head. The PI provided comprehensive information about the sampling procedure to HEWs and district health office heads. HEWs selected community members who lived in the kebeles based on the criteria to participate in the IDI and FGD, while district health office heads selected KII participants.

## Definitions of variables

**Perceptions** are an individual's or group's unique ways of looking at MHS and the barriers and facilitators of MHSU by integrating experiences and memories in the process of comprehending [18].

**Barriers** are obstacles that prevent or block women from utilizing MHS, such as the HFs' physical environment, health care providers (HCPs) attitude and technical ability, lack of

privacy and confidentiality in HFs, long waiting times in HFs, traveling time to reach HFs, availability and accessibility of roads, and prompt transport service [18, 32].

**Facilitators** are promoters that improve or increase the utilization of MHS by providing direct or indirect support, such as the availability and accessibility of HFs and skilled HCPs, essential drugs and supplies, and mothers' knowledge and attitudes towards the existing services [45].

**Mothers or women who were recently delivered** are defined as women who gave birth in the last 12 months preceding the study.

## Data collection procedures

Pre-tested interview guides were utilized to collect data and were adapted from previous similar studies [19–21, 23, 46–48]. The interview guides were prepared in English, translated into the *Sidaamu Afoo* (the main language spoken in the research setting), and then returned to English to ensure consistency and originality. The forward and backward translations were carried out by two separate translators, both English experts and fluent *Sidaamu Afoo* speakers. The translated interview guides were reviewed by the principle investigator (PI) and a third individual who was likewise fluent in both languages. Then, inconsistencies or inaccuracies between the two versions, such as unclear meaning and wording, were addressed based on the assessment.

The PI trained the data collectors for one day before data collection on the importance of the study, data collection processes, aims, methodologies, and ethical considerations. Before data collection, the guides were pre-tested in the Hawela Tula district on participants with similar characteristics and a mix of those involved in the study—the pre-testing aimed to detect ambiguities or deficiencies in the study tool. Thus, important corrections were made based on the feedback after the pre-test.

Three public health experts with MPH degrees, hands-on experience in qualitative data collection, and fluent speakers of *Sidaamu Afoo* collected data using tape recorders. FGD and IDI data were collected using pre-tested interview guides at a suitable place in public areas, whereas KII data were collected at health facilities. Personal or phone contacts were made with all the critical informants before the interview date to arrange the appropriate interview time. Some interviews were conducted after working hours in the interest of the service providers. Open-ended questions were used (see S1 and S2 Files). The open-ended questions were followed by additional probing questions based on the participants' responses. Probing questions bring out more details/information in instances with more clarity or explanations. FGDs were utilized to explore evidence regarding the social context of MHSU and to produce group-level experiences and perceptions by encouraging active interaction. In-depth individual interviews were carried out with participants to scrutinize personal perceptions, some private experiences, and issues, for example, delivery experience and issues raised at the time of FGD that require further probe, as well as to distinguish or identify chances for improving MHS.

The PI ensured the quality of the data by conducting consistent, thoughtful discussions with the data collectors. The discussions were conducted between the interview days to discuss the main results, improve the interview guides, and detect tactics that frequently increase the probe line, following the practice of growing design in the qualitative study. During the data collection, the study subjects were probed to clarify or elaborate on what they had supposed during the group discussions or interviews to check the accuracy and validity of the captured data and the meanings the respondents intended or planned to attribute to it. The transcripts were read several times, recording any information missed during the first readings. A random sample of transcripts was validated against the recordings (A3 File) to facilitate quality assurance.

## Data analysis procedure

After each interview, the data collectors and PI examined the data to improve interview guides and data quality. After the data collection, the tape-recorded data were transcribed verbatim into English. The PI listened to the tape records and read over the transcripts many times to get a general sense of the transcripts and organize the data. Transcripts were exported to MAX-QAD 2020 software for coding and analysis. The transcripts were coded after identifying pre-existing codes from similar studies [19–21, 23, 46–48].

Both inductive and deductive coding methods were utilized. This utilization was guided by pre-established initial codes (open coding) before actual data collection, discussion, and interview guides. Iteratively, throughout reading the transcripts, all transcripts were successively categorized into one of the codes. Extra codes were added while reading the transcripts, categories, and sub-categories that were not formally identified (inductive method). Then, all codes were further analyzed and aggregated into sub-themes and themes (the deductive axial coding approach). The data was used to construct the codes, categories, and themes. We reviewed coding three times when cleaning our themes, categories, and transcripts. The PI carried out line-by-line coding before creating the codebook manual. The PI then accurately coded all of the data. Potential themes were created by clustering categories and codes to answer the study question. The data were analyzed using the thematic content analysis technique by MAXQDA 2020 software. We also utilized the manual analysis method to increase the trustworthiness of the data. We removed unnecessary and irrelevant texts from the *word cloud* before the analysis. The codes were exported from MAXQDA 2020 software to Microsoft Word and Excel to assist in manual analysis.

Data were triangulated from responses from FGDs, IDIs, and KIIs to compare with answers from the various community groups and HCPs. Results from FGDs were triangulated using data from IDIs and KIIs. The concepts and categories that developed from discussions and interviews were confirmed by regularly connecting the emergent classes with the data obtained from the other categories of informants to increase the reliability of the data analysis method. Verbatim quotations were utilized to increase trustworthiness and validate the narrative or story with the respondents' words. Reports of quotes for carefully chosen codes were produced in MAXQAD 2020 software using the code matrix browser. The themes and the arrangement of relationships between the themes were documented and reported. Results were described with the quotations, categories, and significant themes constructed from the data. Finally, results were presented in narratives.

## Ethics statement

All the procedures in this study followed the ethical standards in the Declaration of Helsinki. The Institutional Review Board (IRB) of the College of Medicine and Health Sciences of Hawassa University provided ethical approval under reference number IRB/076/15. The Sidama Region Health Bureau, district health offices, and *kebele* administrators provided letters of support. All community members and HCPs provided written informed consent prior to data collection. Before signing informed written consent, study participants were informed about the purpose of the study, data collection techniques, privacy, voluntary participation, potential benefits, and dangers. Written informed consent was obtained from all study participants before starting the data collection. The confidentiality of the data was ensured during data collection and storage. We ensured the confidentiality of study participants during data collection by interviewing them in a private setting based on their preferences, which adequately maintained their privacy. Besides, we did not collect specific personal identifiers that help identify participants during data storage and transcription. We anonymized all audio files during transcription; only research teams accessed stored files.

## Results

### Characteristics of study respondents

This study included 16 in-depth interviewees, 81 focus group discussants, and 15 key informant interviewees. The 5 IDIs were conducted with recently delivered women, 4 IDIs with WDTs, and 7 IDIs with religious and community leaders. Discussants in FGDs comprised 22 recently delivered women, 14 WDT leaders, 21 community and religious leaders, and 24 *kebeles* leaders. The majority, 13 (61.90%) of community and religious leaders, and 17 (70.83%) of *kebeles* leaders were male among the FGD participants. More than two-thirds 78 (69.64%) of study participants were female. The mean age of focus group discussants was 38.92 years, ranging from 20 to 65. Most (76.54%) focus group discussants had primary education, but few had completed secondary education. The mean age of in-depth interviewees was 30.75 years, ranging from 20 to 39 years. Nine (56.30%) of the in-depth interviewees had primary education. The mean age of key informant interviewees was 27.26 years, ranging from 22 to 39 years. The majority of the key informant interviewees (53.30%) had bachelor's degrees in midwifery and had served for a mean of 5 years as midwives (Table 1).

### Main themes

We distinguished four main themes: practices related to MHS, perceptions of MHS, facilitators of MHSU, and barriers to using MHS. There were several categories under each theme (Table 2).

### Practices of MHS

Nowadays, the majority of mothers use ANC, HFD, child immunization, and modern contraceptive services. Participants stated that the situation of MHS has changed in the present time due to the presence of WDTs and HEWs in *kebeles*, the building of health facilities in their areas, the provision of health education by HEWs and health workers, and other similar initiatives and efforts. A community leader stated this: *"Previously, due to a lack of knowledge, mothers delivered at home, but now, due to the hard work of the WDTs, most mothers are giving birth at the health facilities."* **(FGD, 45-year-old community leader)**

### Why did mothers not go to a health facility before the 45th day after birth?

Participants discussed the value and benefits of obtaining skilled care during pregnancy and labor. They agreed that pregnancy and labor carry the danger of complications as well as a mortality risk. However, neither the mother nor the infant receive proper PNC services. Mothers merely seek medical care after giving birth if they are ill; otherwise, they would wait until 45 days after giving birth to obtain or use family planning and immunization services. A WDT member woman stated this: *"No, women will not go to health facilities before the 45th day after childbirth unless illness happens to them and their children in our area."* **(IDI, 26-year-old woman)**

### Why were mothers not able to complete the recommended number of ANC visits?

The majority of pregnant mothers started ANC visits at four months or later when women could be sure that their pregnancy would continue. One of the WDT members said, *"Mothers will not go there (to health facilities) before four months because their pregnancy is not visible before that time."* **(IDI, 31-year-old woman).** Culturally, mothers did not think that one could

**Table 1. Study participants details.**

| Variables | Categories | Frequency (%) |
|---|---|---|
| Overall participants details | | |
| Mean age | | 36.20 |
| Sex | Male | 34 (30.36) |
| | Female | 78 (69.64) |
| Marrietal status | Single | 11 (9.82) |
| | Married | 101 (90.18) |
| Religion | Protestant Christian | 69 (61.60) |
| | Muslim | 20 (17.86) |
| | Orthodox | 18 (16.07) |
| | Catholic | 5 (4.47) |
| Educational status | Illiterate | 8 (7.15) |
| | Primary education | 71 (63.39) |
| | Secondary education | 12 (10.71) |
| | College diploma and above | 21 (18.75) |
| Occupation status | Housewife | 39 (34.82) |
| | Farmer | 32 (28.57) |
| | Merchant | 20 (17.86) |
| | Government employee | 21 (18.75) |
| In-depth interview particants details | | |
| Mean age | | 30.75 |
| Education status of in-depth interview particants | | |
| | Illiterate | 1 (6.25) |
| | Primary education | 9 (56.25) |
| | College diploma and above | 6 (37.5) |
| Focus groups participants details | | |
| Mean age | | 38.92 |
| Sex of community and religious leaders (n = 21) | Male | 13 (61.90) |
| | Female | 8 (38.10) |
| Sex of *kebeles* leaders (n = 24) | Male | 17 (70.83) |
| | Female | 7 (29.17) |
| Education status of discussant | | |
| | Illiterate | 7 (6.17) |
| | Primary education | 62 (76.54) |
| | Secondary education | 12 (14.82) |
| Key informant interview participants details | | |
| Mean age | | 27.26 |
| Education status of key informant interview | | |
| | College diploma | 7 (46.67) |
| | Bachelor degree | 8 (53.33) |
| Mean service years | | 5 |

know for sure whether their pregnancy would continue before four months. If the pregnancy were to be exposed during the earlier periods, it is thought that it might result in miscarriage. Therefore, admitting to being pregnant is taboo. *"In our area, there is a culture where mothers are not considered properly pregnant before four months. They want to hide their pregnancy by considering it blood or water in their uterus and holding it in secret."* **(KII, 39-year-old midwife)**

**Table 2. Main themes and classes distinguished.**

| Themes | Categories |
|---|---|
| **Practice of maternal health service** | ➢ Maternal health service utilization<br>➢ Frequency and timing of visit |
| **Perceptions of maternal health service utilization** | ➢ Negative<br>➢ Positive |
| **Barriers to antenatal care utilization** | ➢ Lack of awareness of antenatal benefits<br>➢ Distance from health facility<br>➢ Costs associated with MHS use<br>➢ Long waiting time<br>➢ Lack of road access to all-weather road<br>➢ Women being busy with household chores |
| **Barriers to health facility delivery utilization** | ➢ Short or fast on-set labour<br>➢ Lack of an antenatal visit<br>➢ Lack of a birth preparedness plan<br>➢ Non-dignified care<br>➢ Distance from health facility<br>➢ Costs associated with MHS use |
| **Barriers to postnatal care utilization** | ➢ Home delivery<br>➢ Lack of awareness of postnatal care service and schedule<br>➢ Socio-cultural beliefs |
| **Facilitators of maternal health service utilization** | ➢ Previous experience and fear of obstetric complications<br>➢ Health extension workers and women's development teams.<br>➢ Pregnant women's forum |

Most respondents mentioned that women usually visit health facilities twice or thrice and cannot complete the recommended number of visits. A key informant said, *"Women from the better family will come four times. However, the majority of them will come two times after several efforts or pushes following the first contacts."* **(KII, 32-year-old midwife)**

## Perception of and experience with MHS

Most respondents appreciated and were satisfied with the MHS provided by HCPs in health posts, health centers, and hospitals. They also indicated that HCPs were compassionate, caring, and respectful during service provision time at health facilities. A woman who was recently delivered mother stated, *"I think our community is satisfied with the services provided by health professionals in health facilities. They give good care for women during delivery by showing a good face and giving compassionate services, and they assess the health of women and newborn children until the discharge of women."* **(IDI, 25-year-old currently delivered mother).** Also, participants perceived or assured that they had confidence in the services of health facilities. A woman who was recently delivered said, *"We depend on health facilities MHS care and feel that no death will happen there."* **(FGD, 41-year-old currently delivered mother).** Another recently delivered mother woman confirmed, *"Yes, I have confidence in skilled birth attendants' because they have good skills and abilities regarding their work."* **(IDI, 33-year-old currently delivered mother)**

However, few participants reported having negative perceptions of HCPs, delays in receiving care and services, delays in being referred to higher levels of care, specifically while receiving intrapartum care in a health center, and a lack of ambulance service after childbirth. Also, some study respondents reported having had negative experiences with HCPs, health facilities, and ambulance drivers, such as abusive care, a lack of respect, and discriminatory care based on socio-economic level and place of residence. A currently delivered mother who was a

participant in FGD stated that some HCPs' attitudes and behaviors toward mothers are nega-tive. *"I confronted the professionals many times; they have a big attitude problem."* **(FGD, 35-year-old currently delivered mother).** A WDT leader said, *"Sometimes, when we go for ANC service, they are ill-tempered and not welcoming at all. We only go there for the service, not for living, so they have to improve this behavior, and they must provide the service to the mother with sympathy."* **(FGD, 35-year-old woman).** A WDT leader stated, *"The cleanliness of the health center is very poor, and the mosquitoes in the delivery room are causing a problem. Several mothers were discharged early after childbirth due to this problem."* **(FGD, 30-year-old woman)**

Participants noted that women encountered delays in receiving care at the HFs and being referred to the next level of care despite the high perceived need for MHSU. Speaking with the study, respondents revealed negative interactions with the HCPs due to the delay in service delivery. A religious leader said, *"When we take laboring mothers to the health center, they (the HCPs) come late. They always inform us that a long time remains for mothers to deliver, but most mothers deliver immediately. Due to this delay, most mothers developed complications."* **(FGD, 60 years old).** A woman who was recently delivered said, *"The problem I identified is that the HCPs did not refer the laboring mother who needed referral timely; they say we have to watch her for hours."* **(FGD, 30-year-old woman)**

## Barriers to maternal health service utilization

The main barriers to ANC use were lack of awareness of the benefits of ANC, distance from health facilities, costs associated with ANC use, long waiting time, lack of access to roads, par-ticularly in the rainy season, poor knowledge of ANC, and women being busy with different household chores. Distance, costs associated with HFD use, fast on-set labor, lack of an ANC visit, lack of a birth preparedness plan, and non-dignified care were the main barriers to HFD use. The main barriers to PNC use were home delivery, lack of awareness of PNC service and schedule, and socio-cultural beliefs.

### Barriers to antenatal care utilization

**Lack of awareness of ANC benefits.** Participants reported that women who lack aware-ness of the benefits of ANC are less likely to use the service and might not complete the recom-mended number of visits. A key informant said, *"The major reasons are that some women do not know the benefits of ANC service and lack awareness of the negative impacts of not using ANC service."* **(KII, a 25-year-old midwife)**

**Women being busy with different household chores.** Participants reported that women who were busy with different household chores were less likely to use the ANC service and might not complete the recommended number of visits. A key informant said, *"They have family respon-sibilities. For example, those women who have three or more children are highly busy preparing food, caring for younger children, fetching water, keeping animals, etc. Most times, due to these rea-sons, they will not go to HF or become late for an ANC visit."* **(KII, 32-year-old midwife)**

**Long waiting time.** Mothers avoid using ANC due to the long waiting time to obtain ser-vices in health facilities. A key informant affirmed this: *"The long waiting time in the ANC room is due to a shortage of HCPs. . .they (the mothers who went for ANC) will return home without obtaining service if that health professional is attending a delivery. Those women will not come again to the health facility for ANC visits due to these reasons."* **(KII, 32-year-old midwife)**

**Lack of road access.** Most mothers cannot use ANC due to the lack of road access, partic-ularly in the rainy season. A key informant said, *"I have worked in a kebele that doesn't have access to a road, and they (women) may give birth on the road before reaching the health facility."* **(KII, a 30-year-old midwife)**

## Barriers to health facility delivery utilization

**Fast on-set labor.** Most study respondents mentioned that short or fast on-set labor was a barrier that hindered the majority of women from using HFD, even if they planned to give birth in health facilities. A WDTs member said, "*Mostly, mothers will give birth at home due to fast on-set labour.*" **(IDI, 35-year-old woman)**

**Lack of an ANC visit.** Lack of ANC visits during the antepartum period was a major barrier mentioned by the participants as a reason for not using skilled care in the interpartum period. A key informant said, "*Mothers who don't use ANC visits are more likely to deliver at home.*" **(KII, 26-year-old midwife)**

**Lack of a birth preparedness plan.** Most respondents commonly stated mothers' lack of a birth preparedness plan as a reason for not using HFD. A key informant affirmed this: "*Mostly, the mothers who lack a birth preparedness plan will give birth at home in our locality.*" **(KII, 26-year-old midwife)**

**Non-dignified care.** Lack of privacy during the interpartum period was a major barrier mentioned by the participants as a reason for not using skilled HFD at health facilities. A community leader who is an FDG discussant said, "*The mothers prefer to give birth at home because they want to maintain their privacy.*" **(FGD, 32-year-old community leader).** A key informant said, "*More than distance and finance, the non-dignified care will influence them to give birth at home.*" **(KII, 32-year-old midwife).** A WDTs leader affirmed this: "*Older women think that it is not good and unacceptable to open their bodies to health professionals who are younger than them and prefer home delivery.*" **(IDI, 37-year-old woman)**

## Barriers to antenatal care and health facility delivery utilization

**Distance from health facility.** Most mothers could not use MHS due to the distance between their homes and health facilities. "*. . .we have 'Honso or 'Botano' village, which is found beyond the river in a hard-to-reach area. The pregnant woman who carries a fetus in her abdomen from that place cannot access ANC service due to the fact that it requires energy to cross the river and walk the long distance.*" **(FGD, 35-year-old community leader)**. A WDT member mentioned her experience: "*Distance was the factor for me to deliver at home my last child because my labor was short on-set and I did not have enough time to go to a health facility.*" **(IDI, 35 years old)**

**Costs associated with MHS use.** Though the study participants acknowledged and valued the free MHS, there are significant costs associated with MHS use that deter women from seeking MHS. Opportunity costs comprise transport, hidden medical, materials, and food costs for attendees. A key informant affirmed this: "*There are several women who remain at home without using the ANC services due to* direct and indirect costs associated with ANC use. *Thus, poverty is the main barrier to hindering ANC service use.*" **(KII, 32-year-old midwife)**. All participants noted that ambulances would not offer round-trip packages during intrapartum care, and the community suffers from the lack of transportation fees after discharge from the health facility. A woman who was recently delivered said, "*I gave birth to twins in a public hospital after several referral processes. They did not provide me with ambulance service from the hospital to my home after discharge. I used public transportation to come back to my home.*" **(FGD, 27-year-old woman)**

## Barriers to postnatal care utilization

**Home delivery.** The uptake of MHS across the continuum is impacted by the use of prenatal and interpartum care. Most women were discouraged from using skilled PNC from the health facility after home delivery. Participants said women do not get the proper PNC service

after home delivery. A WDTs leader affirmed this: *"Mothers miss PNC service if they don't deliver at a facility"* **(IDI, 26-year-old woman)**

**Lack of awareness of PNC service and schedule.** Most respondents mentioned that women do not obtain PNC services after childbirth unless they encounter health problems or their children are sick before the 45[th] day of childbirth. They claimed that PNC service is required solely in cases where women experience complications or illness. *"Women will go to health facilities 45 days after delivery to get family planning for themselves and immunizations for their children. She doesn't go before the 45[th] day after delivery."* **(IDI, 33-year-old currently delivered woman)**. Most respondents did not know about PNC services, and they mentioned PNC as just equating with postpartum family planning and immunization services that women obtained at or after 45 days of childbirth. A key informant said, *"If the mother gave birth in our health facility, we discharge them between six and twenty-four hours later due to a shortage of beds and rooms. However, they will come again on the 45[th] day for child vaccination and family planning services unless they have experienced health problems."* **(KII, 32-years-old midwife)**

**Socio-cultural beliefs.** Some participants mentioned that women are not using the PNC service after childbirth due to socio-cultural barriers. A key informant said, *"Old people will prevent mothers from going outside the home due to socio-cultural reasons and attitudinal problems. Most people think the women will be exposed to 'mich' or' buda' (evil eye) if they go outside their home during the postpartum period."* **(KII, a 25-year-old midwife)**

**Facilitators of maternal health service utilization.** Previous experience and fear of obstetric complications, health extension workers and women's development teams, and pregnant women's forums were the main facilitators of maternal health service utilization.

**Previous experience and fear of obstetric complications.** Most participants stated that previous experience and fear of obstetric complications and suspicion of their recurrence influenced mothers to use MHS. A key informant said, *"Most times, there is an event that motivates them to come here. For instance, I think in 2014 E.C. (2021/22), a woman was delayed at home for long times after labour started due to a prophetic command to give birth at home and developed serious complications. They brought the woman here, and we referred her to a nearby primary hospital. Then, the nearby primary hospital also referred her to a referral hospital, but the woman died there. If they hear about this kind of event, all of them will come to a health facility for institutional delivery care."* **(KII, 32-year-old midwife)**

**Health extension workers and women's development teams.** Participants mentioned that HCPs, particularly HEWs, motivated the women to use MHS. The WDTs motivated laboring mothers by reporting to HEWs and dialing for ambulance services. Also, they showed that the WDTs transported the mother to an ambulance arrival point or health facilities and back to their houses, specifically when there was a lack of transport services. A woman who was recently delivered mother said, *"WDTs and HEWs are motivating us to get services from health facilities."* **(FGD, 27-year-old woman)**. A key informant stated this: **"***They (WDTs) bring a laboring mother to our facility. Also, they call ambulances when there is a need for them. They also report to us whenever home delivery occurs. I think they are helping with service delivery in our area."* **(KII, 24-year-old midwife)**

**Presence of a pregnant women's forum.** Most participants stated that the pregnant women's forum motivated the women to use MHS at health facilities. A key informant affirmed this: *"Most times, the pregnant women's forum motivates them to attend health facility deliveries."* **(KII, 32-year-old midwife)**

## Discussion

We explored the perceptions of maternal health service and barriers and facilitators of maternal health service use in the Sidama region of southern Ethiopia. Results indicate that

communities have positive perceptions and good practices about skilled ANC and HFD, but the majority of mothers do not use care during the postpartum period. Some participants experienced negative interactions with HCPs, health facilities, and ambulance drivers, such as abusive care, a lack of respect, and discriminatory care based on socio-economic level and place of residence; delays in receiving care and services; delays in being referred to higher levels of care, specifically while receiving intrapartum care in health centers; and a lack of ambulance service after childbirth.

The main barriers to ANC use were lack of awareness of ANC benefits, distance from health facilities, costs associated with ANC use, long waiting times, lack of road access, and women being busy with different household chores. Distance from the health facility, costs associated with HFD use, fast on-set labor, lack of an ANC visit, lack of a birth preparedness plan, and non-dignified care were the main barriers to HFD use. The main barriers to PNC use were home delivery, lack of awareness of PNC service and schedule, and socio-cultural beliefs. The main identified facilitators of MHS use were previous experience and fear of obstetric complications, health extension workers and women's development teams, and pregnant women's forums.

Most of the community members have a positive perception and good practice of skilled ANC and HFD, but a significant number of women dropped out of receiving skilled care during the postpartum period, which is in agreement with the 2019 Mini-EDHS report, where about three-fourths, half, and one-third of women had at least one ANC visit, HFD care, and PNC service, respectively [13]. Also, the Mini-EDHS showed a two- and five-fold increase in skilled ANC and HFD but an insignificant change in PNC in the last decade between 2011 and 2019 [6, 13]. The results show that the MHSU has improved through different initiatives and efforts. First, the government of Ethiopia reorganized community engagement in 2011, and the WDA strategy was created to improve the health extension program further. The WDA members assist HEWs in spreading essential messages about skilled MHS via social events, including coffee ceremonies, peer-to-peer marketing, and other neighborhood events. They detect pregnant mothers and mothers with term pregnancies in their neighborhoods and connect them with HEWs for early ANC and HFD services [46]. Second, training and deployment of HEWs and HCPs, particularly midwives in rural settings; expansion of health facilities; introduction of ambulance service; and the provision of MHS free of charge [29] have helped improve MHSU. However, in the current study area, the presence of socio-cultural beliefs among residents that movement outside the home may expose women to evil spirits may decrease PNC utilization by restricting the travel of women after giving birth. Researchers argued that women from rural communities in Ethiopia had been challenged to use PNC due to socio-cultural beliefs. Similar findings were documented in studies conducted in different settings [48, 49].

Lack of awareness of the benefits of ANC was a barrier to ANC service use and being unable to complete the recommended number of visits. Similar results were reported from the studies conducted in the Sidama region of south Ethiopia [19], Indonesia [21], and south Sudan [20]. The likely explanation is that women who know the ANC service tend to have a good understanding of ODS, a positive attitude, health-seeking behaviors, and the confidence to use the service.

The long distances to health facilities and lack of road access were the main barriers to ANC and HFD use. The inaccessibility of MHS due to a lack of road access and distance remain significant barriers to obtaining MHS in Ethiopia, regardless of free MHS, free ambulance service from houses to facilities, and expansion of primary healthcare to assure universal access to primary healthcare [29]. This result agreed with studies done in the Tigray region of Ethiopia [22], Indonesia [21], South Sudan [20], and Thailand [23]. Long distances and poor

road access are the primary geographic barriers for pregnant women in Ethiopia seeking the services of skilled HCPs. Maternity waiting homes provide a pathway for women to deliver in health facilities to address this issue, thereby contributing to the decrease of the alarming maternal death trend and lousy pregnancy outcomes. However, maternal waiting home utilization is low in Ethiopia, and the government needs to exert more effort to promote its utilization, particularly in rural settings [50].

This study also identified costs associated with MHS use as barriers to ANC and HFD use. Researchers argued that women from resource-constrained communities had been challenged to pay for healthcare, and these costs posed economic barriers to utilizing MHS [24, 25]. Therefore, due to a lack of economic access, the mothers may not visit ANC at all, decrease the number of recommended ANC follow-ups, or even start ANC in late pregnancy and give birth at home. Due to increased out-of-pocket expenses for transport, home return transportation costs, medical care, and food, the utilization of services is hampered [21]. Through community-based outreach services, HF expansion, health insurance packages, and voucher programs, there is still a need to reach more women for quality care [51].

Women being busy with different household chores was another main barrier preventing mothers from using ANC services. Ethiopian women generally share a huge household workload, which is very high in rural settings [52]. Traditional male roles in our societies restrict male participation in household chores [53, 54]. For example, while women become increasingly involved in financially providing for their families, their male spouses are not increasing their participation in child care and domestic activities s [54]. This disparity upholds the cultural standard of the mother being primarily accountable for all domestic activities in addition to her outside work. [53]. Even though men and women in this country realize that males can perform traditionally female jobs, customs remain that men should not take part in household tasks [54]. This exhausts and overwhelms most women of childbearing age, which hampers their ability to go to HF at all or become late for an ANC visit [55]. This result agrees with studies done in Hossaina town in Ethiopia [31] and South Sudan [20].

This study found that long waiting times in ANC rooms are a barrier to using ANC services. The long waiting time at the health facility may cause direct and indirect costs for women. The direct costs comprise money for food, transport, and time spent. The indirect costs comprise the responsibility left uncharged for each ANC visit appointment day. These responsibilities comprise childcare, formal or informal service, and other house jobs. While some mothers rely on friends and family to help with childcare and domestic chores, others must choose between those duties and their ANC visit. The long wait times raise these costs, making it more difficult for mothers to attend ANC. A similar result was documented in a study in southern Mozambique [26].

This study found that fast on-set labor is the main barrier to HFD use. The main obstacle to HFD was found to be labor's on-set unpredictability. The majority of labors were reported to start at night, as was to be expected, during the time of transportation inaccessibility to the HF, "forcing" the mothers to give birth at home. Maternity waiting homes in Ethiopia are one of the strategies being used to overcome this barrier by accommodating women in their final weeks of pregnancy and addressing the geographical gap in obstetric care. However, utilization of maternity waiting homes among pregnant women was meager in Ethiopia, including the southern part [56–58].

Most previous studies [20–23] that reported long distances from a health facility and the cost of transportation as barriers to HFD have not made a clear relationship between these factors and the prediction of labor on-set, leading to narrowly focused interventions that merely address the costs of transportation. Our findings point to the necessity of making transportation arrangements in advance for the HFD use during unpredictable labor on-set. A birth plan

incorporating transportation planning is currently one of the ANC interventions. However, this intervention does not fully alleviate the problem, even for women who attend ANC. HCPs are using women's memories of their most recent period to estimate the expected date of delivery (EDD), which is frequently incorrect and results in incorrect EDD forecasts. Women who had previously had ANC claimed that their deliveries had taken place much earlier or later than the proposed dates that had been given to them by the HF. Therefore, finding better methods of estimating EDD could be a way to influence or intervene in HFD use. Better EDD estimation approaches, such as early obstetric ultrasound, manual physical examination with fingers, and last menstrual period for women with regular menstrual cycles, are relevant in Ethiopia. Obstetric ultrasound is now available in all primary hospitals and urban health centers but not all rural health centers, limiting its utilization in ANC clinics.

The current study identified that the lack of an ANC visit is the main barrier to HFD use. This result aligns with other studies that reported a lack of ANC visits as a strong barrier to HFD use [17, 47]. Numerous studies have found an association between ANC visits and health facility delivery, but these studies have not explored why this is so in-depth. Most studies have been cross-sectional in design and heavily depend on information from house surveys [17, 33, 34]. Our study extends the results of these earlier studies by revealing the reasons for the low attendance we observed and its effects on subsequent visits. Our findings can help explain this association in light of the fear of HCPs' criticism for skipping ANC. This implies that health professionals' attitudes need to change. At any point in their maternity periods, the women should be allowed to access the HF care system without being shamed or turned away for not having visited earlier. Because of the experience shared by women FGD discussants, after negative interactions with HCPs due to the skip of ANC, they then decide to deliver at home and influence others to do so. Thus, it is also necessary to change the negative attitude of HCPs toward women who choose to use MHS at any stage of the continuum of care.

The result of this study highlighted the lack of a birth preparedness plan as a significant barrier to HFD use. One explanation for this could be that well-prepared women better understand ODS and effective communication skills with HCPs. As a result, they might have made all the necessary preparations to use HFD services effectively and efficiently. According to other studies, women who are knowledgeable about ODS are more likely to be prepared for childbirth, be aware of potential difficulties, and frequently utilize skilled care from HFs [59–61].

The results of this study noted non-dignified or disrespectful care as a main barrier to women's use of HFD. This has an echo effect that hampers women from using interapartum care in health facilities. Earlier studies conducted in Ethiopia [62] and elsewhere [63] revealed that mothers experienced different types of maltreatment in health facilities. According to the Ethiopian GTP, patient-centered, compassionate, and respectful care is a top priority in efforts to increase service quality and equity. The Ethiopian government has been exerting efforts at all health facilities to possess caring, respectful, and compassionate health professionals [27]. It is an approach centered on the individual based on principles of ethics (including respect for women's autonomy, dignity, feelings, choices, and preferences) and respect for human rights that promotes practices that recognize women's needs [64]. Even though several factors contribute to low MHSU, it is becoming clear that provider abuse is among the reasons that many women are unable to seek MHS [65]. Several studies have found that women's views of how they will be treated at healthcare facilities may have a significant impact on where they want to get MHS, particularly in childbirth [66, 67].

This study found that home delivery is a barrier to using PNC services, which agrees with a study done in rural Kenya [47]. This finding can be partly explained by the evidence that HFD is one of the vital linkages between women and HCPs. The women who have HFD would have a high probability of receiving adequate counseling and information on ODS and the

importance of skilled PNC. Also, mothers seeking health care throughout their pregnancy and childbirth might be more likely to seek it during their postpartum period. Moreover, the mothers who regularly visit HFs for ANC and HF use have earlier indicated their acceptance of the health system.

Lack of awareness of the PNC service and schedule among community members, particularly women, about the availability and benefits of the service, is another main barrier to its uptake. Participants felt that PNC was only necessary in the event of ill health and complained that they were never told to return for PNC after delivery. Similarly, it has been noted elsewhere [21] that community members do not comprehend the significance of MHS services, particularly in the postpartum period. This demonstrates how the continuum of care for women's antepartum and intrapartum encounters with the health system is seriously lacking in Ethiopia, where maternal health services are fragmented [68]. For Ethiopia, where almost half of the expected three million yearly birth cohorts occur at home, a mixed-method service provision modality that includes home and facility-based PNC services and home visits by HCPs or community workers may be beneficial [13].

Regardless of the barriers, this study documented facilitators of MHS in the study area. These enablers should be considered to increase MHS service provision. These comprise previous experience and fear of obstetric complications, the efforts of HEWs and WDTs, and a pregnant women's forum.

Previous experience and fear of obstetric complications motivated women to use MHS, which is in line with previous studies' findings from Debre Markos town [32] in Ethiopia and Nepal [69]. The most plausible explanation is that being exposed to complications raises women's and families' fears of having the same problem again. Furthermore, women who have observed major warning signals are more likely to have perceptions of vulnerability and the severity of dangers, which directly lead to increased MHSU. Another factor could be women's understanding of ODS, which could be a strong drive for the mother to seek MHSU as soon as difficulties arise.

HEWs and WDTs motivated women to use MHS, which agrees with other studies that revealed the HEWs and WDTs structure at the community level strongly contributed toward MHSU [70–72]. The possible justification could be that women are expected to actively participate in the one-to-five networks below the WDTs at the community level. This allows them to quickly access primary healthcare facilities for information, care, and support. Another important motivator for MHSU is the pregnant women's forum. The reasons would be that mothers who participated in the pregnant women forum had more focused counseling, good knowledge of ODS, skills in birth planning, an ambulance driver's phone number, good communications with HEWs and HCPs, a positive attitude, and good health-seeking behaviors.

## Limitations of the study

There were some limitations to this study. First, the findings might be susceptible to recall and social desirability biases because the data were collected from study participants' self-reports. Study subjects might be unable to recall most of the barriers and facilitators of MHSU, which may affect their link with MHSU. There is the risk of purposely misquoting personally related perceptions, barriers, facilitators, women's attitudes towards MHS, perceived quality of care, and interaction with HCPs and HFs (social desirability bias). Thus, the degree of these factors might have been overvalued or undervalued, and as such, the link between these factors and MHSU might have been overestimated or underestimated. A selection bias might be likely for this study because HEWs selected community members. The HEWs might invite interested study participants to participate. As a result, the respondents might have more favorable

perceptions and good practices toward MHSU. Another issue is that qualitative results are subjective and influenced by the individual's surroundings, making conclusions and comparisons difficult.

Despite these limitations, our study has several strengths. Data triangulation was used to assure the data's trustworthiness; the data were collected from several sources, including recently delivered women, WDT leaders, community and religious leaders, *kebeles* leaders, and health care providers. Also, as methodological triangulation, this study used IDIs, KIIs, and FGDs to collect data on the same topic being examined. To ensure the data's credibility, we designed the open-ended queries to be transparent, non-leading, and impartial to avoid bias throughout the research procedure. Data collectors collected the data with prior experience in qualitative data collection, and we retained neutrality during the data collection process so as not to influence the respondents' answers. To minimize bias and data misinterpretation, we asked and probed the respondents to explain or elaborate on what they had said during the group discussions or interviews; we also reviewed the results and meanings of the data. Besides, data collection procedures, notes of any field decisions, analysis notes, raw data, and data interpretation were thoroughly documented to assure data confirmability. Finally, we comprised more individuals with different viewpoints to get a complete picture of MHSU.

## Conclusions

Rural women in southern Ethiopia still experience several challenges while using MHS. The health care provision structure at different levels does not comprehensively address mothers' desires due to disrespectful and unfriendly HCPs, abusive care, discriminatory care based on a socio-economic level and place of residence, long waiting times, a lack of urgent, timely referral, high direct and indirect costs associated with MHS use, distance from the health facility, a lack of road access, and transportation arrangements during unpredictable labor. Any intervention programs should address these barriers that continue to impede several mothers from using MHS. Also, specific intervention strategies should be designed for women with poor awareness of MHS, mothers who are busy with different domestic chores, and mothers who missed using MHS earlier. Furthermore, PNC is poorly executed in the study area. Therefore, it is necessary to implement and strengthen the provision of quality PNC services based on World Health Organization guidelines and country directives. Moreover, there is an urgent need to educate communities to circumvent socio-cultural beliefs that hinder postpartum health care use.

## Supporting information

**S1 File. English version interview guide.**
(DOCX)

**S2 File. Sidaamu *afoo* version interview guide.**
(DOCX)

**S3 File. Transcript.**
(DOCX)

## Acknowledgments

We are very thankful to the study participants, data collectors, supervisors, and administrators at different levels in the Sidama region who directly and indirectly contributed to this study.

Lastly, our superior thanks go to Netsanet Kibru for her immense support, such as the duplication of the consent form.

## Author Contributions

**Conceptualization:** Amanuel Yoseph, Ayalew Astatkie.

**Data curation:** Amanuel Yoseph, Ayalew Astatkie.

**Formal analysis:** Amanuel Yoseph, Ayalew Astatkie.

**Funding acquisition:** Amanuel Yoseph, Ayalew Astatkie.

**Investigation:** Amanuel Yoseph, Wondwosen Teklesilasie, Francisco Guillen-Grima, Ayalew Astatkie.

**Methodology:** Amanuel Yoseph, Wondwosen Teklesilasie, Francisco Guillen-Grima, Ayalew Astatkie.

**Project administration:** Amanuel Yoseph, Wondwosen Teklesilasie, Francisco Guillen-Grima, Ayalew Astatkie.

**Resources:** Amanuel Yoseph, Ayalew Astatkie.

**Software:** Amanuel Yoseph, Ayalew Astatkie.

**Supervision:** Amanuel Yoseph, Ayalew Astatkie.

**Validation:** Amanuel Yoseph, Wondwosen Teklesilasie, Francisco Guillen-Grima, Ayalew Astatkie.

**Visualization:** Amanuel Yoseph, Wondwosen Teklesilasie, Francisco Guillen-Grima, Ayalew Astatkie.

**Writing – original draft:** Amanuel Yoseph, Wondwosen Teklesilasie, Francisco Guillen-Grima, Ayalew Astatkie.

**Writing – review & editing:** Amanuel Yoseph, Wondwosen Teklesilasie, Francisco Guillen-Grima, Ayalew Astatkie.

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
