## [Decision Letter · Decision Letter 0]

26 Jun 2024

PONE-D-24-12696Perceptions, barriers, and facilitators of maternal health service utilization in southern Ethiopia: A qualitative exploration of community members’ and health care providers’ viewsPLOS ONE

Dear Dr. Yoseph Samago,

Thank you for submitting your manuscript to PLOS ONE. After careful consideration, we feel that it has merit but does not fully meet PLOS ONE’s publication criteria as it currently stands. Therefore, we invite you to submit a revised version of the manuscript that addresses the points raised during the review process.

We look forward to receiving your revised manuscript.

Kind regards,

Jackline Oluoch-Aridi, Ph.D.

Academic Editor

PLOS ONE

2. Please amend the manuscript submission data (via Edit Submission) to include author Amanuel Yoseph .

3.  Please amend your authorship list in your manuscript file to include author Amanuel Yoseph Samago.

4 Please review your reference list to ensure that it is complete and correct. If you have cited papers that have been retracted, please include the rationale for doing so in the manuscript text, or remove these references and replace them with relevant current references. Any changes to the reference list should be mentioned in the rebuttal letter that accompanies your revised manuscript. If you need to cite a retracted article, indicate the article’s retracted status in the References list and also include a citation and full reference for the retraction notice.

Additional Editor Comments (if provided):

Reviewers' comments:

Reviewer's Responses to Questions

**Comments to the Author**

1. Is the manuscript technically sound, and do the data support the conclusions?

Reviewer #1: Yes

Reviewer #2: Yes

2. Has the statistical analysis been performed appropriately and rigorously? 

Reviewer #1: N/A

Reviewer #2: Yes

3. Have the authors made all data underlying the findings in their manuscript fully available?

Reviewer #1: Yes

Reviewer #2: Yes

4. Is the manuscript presented in an intelligible fashion and written in standard English?

Reviewer #1: Yes

Reviewer #2: Yes

5. Review Comments to the Author

Reviewer #1: This manuscript addresses maternal health services utilization in Southern Ethiopia. Overall, the manuscript is well written and provides important contribution to the field. Some recommendations are provided below to improve readability and application of the findings for program managers

INTRODUCTION

Line 73-77 – The authors state that great differences in maternal mortality exist across regional states. Given that the study was conducted in the Southern region, authors should indicate the prevalence of maternal mortality in the Southern region.

Line 88-89 “Nonetheless, the utilization of the existing MHS is low in developing countries, particularly SSA [2]” The cited reference is the World Health Organization (2023) Malnutrition trend in developing countries. Lancet 32: 794 10. – yet the sentence speaks to maternal health service utilization in developing countries. A more appropriate reference should be given as well as estimates for the low utilization of maternal health services in developing countries to justify the statement.

METHODOLOGY

- Give a brief summary of the number of regions in Ethiopia for the reader unfamiliar to the area.

- What is the approximate population in ethiopia?

- The inclusion and exclusion criteria is not clear - Clearly indicate the inclusion and exclusion criteria for the FGDs, IDIs and KII’s

- Provide an explanation for the choice of the groups selected for the IDIs and FGDs including a brief explanation on who are WDTs. Are there differences between the WDAs previously discussed in the introduction and WDTs?

- Indicate where the FGDs, IDIs and KIIs were held.

- Line 277-278 states “ Iteratively, over the course of reading the transcripts, all transcripts were successively categorized into one of the codes”, suggesting that entire transcripts were categorized into a single code, rather than segments of the transcripts being coded. The statement should be revised e.g., "Iteratively, over the course of reading the transcripts, all segments of the transcripts were successively categorized into various codes."

- Some of the information provided in the methodology can be summarized to make the section more concise e.g., the sample size determination section Line 179-186

- Were separate FGDs held between women and male groups?

RESULTS

Provide details on the gender of the WDT leaders, community, religious and kebele leaders.

To improve readability and comprehensibility, the authors should group the MHSU barriers based on the specific maternal health services as defined in the introductory section of the barriers to MHSU i.e. barriers to ANC use, barriers to HFD use, barriers to PNC use and cross-cutting barriers and discuss the specific barriers under each sub-section.

DISCUSSION

Line 565 is incomplete– “This result agreed with the studies done in the…..”

Line 561- 565 and 677-681 both discuss socio-cultural factors that impede utilization of PNC services and should be merged to make the discussion more concise. Given that this finding has been highlighted in previous studies, are there any interventions that have been tried to address this or what recommendations would the authors propose for program managers?

Line 618-622 - The authors highlight fast onset labour and distance from facilities as key barriers to HFD. One of the interventions in Ethiopia to try and overcome this barrier are maternity waiting homes that accommodate women in their final weeks of pregnancy to bridge the geographical gap in obstetric care. Could the authors provide more information on the use of this and such interventions with relation to the studied region.

Line. 621-622 -The authors relate the use of LMP to estimate EDD as a reason for inaccurate EDDs leading to unpredictable delivery dates. They suggest better methods for estimating EDD as a measure to influence HFD. Could the authors briefly provide examples of the suggested methods and their applicability in the Ethiopian context.

GENERAL COMMENTS

Compare Line 86 and 755 defining MHSU. Authors should stick to one MHSU abbreviation definition and not interchange between maternal health service use and Maternal Health Service Utilization.

Line 105 – Edit the statement to “universal inaccessibility of maternal health services”

Line 229 – Delete the word maintain

Line 320 – Delete the space after discussants

Line 587 – Remove the apostrophe and letter ‘s’ from the word women’s

Reviewer #2: Congratulations to the authors for this piece of work. I have the following comments that the authors should consider.

In the introduction section, provide some details about MHS being free. This is mentioned in the results but not in the background. This will help the reader with understanding the context better.

Sample size determination:

There is repetition in the sample size determination for IDIs. This is mentioned in lines 168 - 172 and again in lines 186 - 191. Secondly, in the first instance the authors say they settled on 16 participants but this number changes to 15 in the second instance. Review this and align.

Provide more clarity on the difference between IDIs and KIIs and who they were conducted with.

In line 249 the authors mention that probing questions were asked to capture experiences and perceptions. This can be understood as meaning that only the probing questions were aimed at bringing these out while the study is about experiences and perceptions and as such all questions would be aimed at this. The authors can rephrase this. Probing questions are used to bring out more details/information in instances where more clarity or explanations are needed.

Ethics statement:

In lines 311 - 312, explain how confidentiality was ensured.

Results:

Lines 343 - 349 and 361 - 365: If the reasons for these two findings were explored, than that needs to come out clearly. why did mothers not go to a health facility before the 45th day after birth? Why were mothers not able to complete the recommended number of ANC visits?

Lines 425 - 430: the authors mention of free MHS and also that there were medical costs. Why were there medical costs if the services were free? Is this finding about indirect costs only? This should come out clearly. If there were medical costs with services being free, then the reason for this should be explained. The authors can bring out the effect of indirect or non-medical costs more clearly.

Lines 465 - 468: Is there an explanation as to why there was lack of privacy? Is it overcrowding? inadequate infrastructure? A key objective of the study should be to explain the reasons why things are happening the way they are.

Line 565 second sentence seems incomplete.

Lines 572 - 577: The authors need to bring out the significance of transportation costs in a context of limited geographical access and cite relevant literature. This is a significant barrier to access to services and needs more emphasis.

Lines 578 - 586: the authors should explain why women are incurring medical costs when MHS are free. See my earlier comment.

Discussion:

While the authors have done well to summarize their findings, they have repeated a lot of their findings here. They can summarize the key findings in a paragraph and focus more on situating their findings in wider literature which they have done to some extent but more can be done.

6. PLOS authors have the option to publish the peer review history of their article (what does this mean?). If published, this will include your full peer review and any attached files.

Reviewer #1: No

Reviewer #2: No

---

## [Author Response · Author response to Decision Letter 0]

2 Aug 2024

Point-by-point responses to reviewers’ and editor’s comments

Reviewer 1

Comment 1: This manuscript addresses maternal health services utilization in Southern Ethiopia. Overall, the manuscript is well written and provides important contribution to the field. Some recommendations are provided below to improve readability and application of the findings for program managers.

Authors’ response: Thank you for your kind remark. 

Comment 2: Line 73-77 – The authors state that great differences in maternal mortality exist across regional states. Given that the study was conducted in the Southern region, authors should indicate the prevalence of maternal mortality in the Southern region. 

Authors’ response: Thank you a lot for this valuable comment. We have accepted the comment and made the required revision. The introduction has been edited to be more logical and coherent by including maternal mortality data as per your suggestion.

Comment 3: Line 88-89 “Nonetheless, the utilization of the existing MHS is low in developing countries, particularly SSA [2]” The cited reference is the World Health Organization (2023) Malnutrition trend in developing countries. Lancet 32: 794 10. – yet the sentence speaks to maternal health service utilization in developing countries. A more appropriate reference should be given as well as estimates for the low utilization of maternal health services in developing countries to justify the statement.

Authors’ response: Thank you for this important comment. We have revised the references section to avoid ambiguity, as indicated below in journal requirement 4, and this references section error was unintentionally introduced because we used Endnote referencing manager software. Now we have fixed it and provided estimates of maternal health service utilization to justify our statements as per your suggestion. 

Comment 4: Give a brief summary of the number of regions in Ethiopia for the reader unfamiliar to the area.

Authors’ response: Thank you for the genuine and plausible comment. The Federal Democratic Republic of Ethiopia currently consists of 13 regional states. Dear reviewer, we kindly request that you look at the study area part of this paper. Now we have provided the required information as per your suggestion to make the study area part of this manuscript clearer for an unfamiliar reader.

Comment 5: What is the approximate population in Ethiopia?

Authors’ response: Thank you for pointing this out. As indicated in the above comment, we have now provided the approximate population size of the country. 

Comment 6: The inclusion and exclusion criteria is not clear - Clearly indicate the inclusion and exclusion criteria for the FGDs, IDIs and KII’s

Authors’ response: Thank you for this comment, too. We have clearly indicated the inclusion and exclusion criteria for the FGDs, IDIs, and KIIs in the “study design and population” part of the methods based on your suggestion.

Comment 7: Provide an explanation for the choice of the groups selected for the IDIs and FGDs including a brief explanation on who are WDTs. Are there differences between the WDAs previously discussed in the introduction and WDTs? 

Authors’ response: Dear reviewer, thank you for your important comment. We have accepted comments and provided sufficient explanation about the IDIs and FGDs participants’ selection procedures. Besides, these two terms are similar at the structure level in Ethiopia, which consists of 30 women in one group. However, currently, the Ethiopian government has changed the name of the community structure from the Women Development Army (WDA) to the Women Development Team (WDT). We have used the term WDA to indicate the Ethiopian government's effort to reorganize community structures to increase maternal health service utilization during the previous period. In the method parts, we have used the currently utilized term to become consistent with the government-updated term.

Comment 8: Indicate where the FGDs, IDIs and KIIs were held.

Authors’ response: Thank you for this important comment. However, we have already indicated a place where the FGDs, IDIs, and KIIs were conducted in the “data collection procedure” part of the manuscript. FGD and IDI data were collected using pretested interview guides at a suitable place in public areas, whereas KII data were collected at health facilities.

Comment 9: Line 277-278 states “Iteratively, over the course of reading the transcripts, all transcripts were successively categorized into one of the codes”, suggesting that entire transcripts were categorized into a single code, rather than segments of the transcripts being coded. The statement should be revised e.g., "Iteratively, over the course of reading the transcripts, all segments of the transcripts were successively categorized into various codes."

Authors’ response: Thank you for pointing this out and for fixing some of the typographic and grammatical errors we committed. The language has now been carefully and extensively edited by the authors and by a colleague, namely Ashenafi Bogale, who is an assistant professor of English language at Hawassa University.

Comment 10: Some of the information provided in the methodology can be summarized to make the section more concise e.g., the sample size determination section Line 179-186. 

Authors’ response: Thank you for this comment. We have accepted your comment and did the required revisions as per your comment to make the make the section more concise. 

Comment 11: Were separate FGDs held between women and male groups?

Authors’ response: Thank you for this query. We have conducted separate FGDs among three groups. The first group was women who gave birth in the 12 months preceding the study, which only consisted of women. The second and third groups were community and religious leaders and kebele leaders, which consisted of both male and female study participants regardless of any exclusion criteria. However, we have selected them using a maximum variance sampling method to ensure sufficient heterogeneity. Dear reviewer, kindly look at the “sample size determination and sampling procedure” section of methods.

Comment 12: Provide details on the gender of the WDT leaders, community, religious and kebele leaders.

Authors’ response: Dear reviewer, thank you for your critical comment. However, there were no male WDT leaders in the study area due to the very nature of the structure, which consists of only women. Now we have provided the overall gender mix of this study, particularly community, religious, and kebele leaders’ gender details, as per your suggestion to make the results more visible for readers.

Comment 13: To improve readability and comprehensibility, the authors should group the MHSU barriers based on the specific maternal health services as defined in the introductory section of the barriers to MHSU i.e. barriers to ANC use, barriers to HFD use, barriers to PNC use and cross-cutting barriers and discuss the specific barriers under each sub-section. 

Authors’ response: Thank you for this comment, too. We have cautiously and comprehensively revised this section to improve readability and understandability, as per your comment.

Comment 14: Line 565 is incomplete– “This result agreed with the studies done in the…..”

Authors’ response: Thank you for this important comment. We have now included a description to complete the sentence.

Comment 15: Line 561- 565 and 677-681 both discuss socio-cultural factors that impede utilization of PNC services and should be merged to make the discussion more concise. Given that this finding has been highlighted in previous studies, are there any interventions that have been tried to address this or what recommendations would the authors propose for program managers?

Authors’ response: Thank you. We have merged the discussion of our findings as per your comment to make the discussion more concise. As you correctly pointed out, we have developed a community-based health education intervention (HEI) that focuses on knowledge of obstetric danger signs, birth preparedness and complication readiness practices, and barriers to MHSU. In light of this, we have tried to address all barriers to MHSU in our intervention, but we have not particularly addressed socio-cultural barriers because our intervention was designed before the analysis of qualitative findings. Yet, that wouldn’t exclude the possibility of social-cultural beliefs contributing to the very low utilization of PNC in study settings. Our project consisted of four papers, two of which were published in SAGE Women’s Health and MDPI Healthcare Journals. We have addressed this issue as a possible limitation of the study in the discussion section of Paper IV. Our six-month community-based HEI significantly increased the utilization of skilled ANC and HFD but did not improve the utilization of PNC. Thus, expanding the HEI with certain modifications, for instance, mobilizing more stable and active community members, addressing demand and supply-side concerns related to distance from health facilities, costs associated with MHSU, waiting time to obtain MHS, road accessibility, the transportation arrangements during unpredictable labor, the needs of poor mothers, sociocultural barriers, quality of services, and skilled HCPs, as well as repeated or longer HEI, will benefit in attaining a superior effect in improving the utilization of MHS. Furthermore, PNC utilization is very low during the early postnatal period; adaptation of HEI must be prioritized, and attention given to the inclusion of husbands, more socio-culturally adherent community groups about postpartum taboos of hiding delivered mothers, and home-based visits should be considered to increase PNC utilization. Dear reviewer, kindly look at the link to the published Paper IV for further information. Yoseph, A.; Teklesilasie, W.; Guillen-Grima, F.; Astatkie, A. Community-Based Health Education Led by Women’s Groups Significantly Improved Maternal Health Service Utilization in Southern Ethiopia: A Cluster Randomized Controlled Trial. Healthcare 2024, 12, 1045. https://doi.org/10.3390/healthcare12101045

We also suggested, particularly for this paper, an urgent need to educate communities to circumvent socio-cultural beliefs that hinder the use postpartum health care (see last paragraph of the conclusion section).

Comment 16: Line 618-622 - The authors highlight fast onset labour and distance from facilities as key barriers to HFD. One of the interventions in Ethiopia to try and overcome this barriers are maternity waiting homes that accommodate women in their final weeks of pregnancy to bridge the geographical gap in obstetric care. Could the authors provide more information on the use of this and such interventions with relation to the studied region? 

Authors’ response: Thank you for the important comment. We have revised this section of discussion and provided additional details as per your suggestion. 

Comment 17: Line. 621-622 -The authors relate the use of LMP to estimate EDD as a reason for inaccurate EDDs leading to unpredictable delivery dates. They suggest better methods for estimating EDD as a measure to influence HFD. Could the authors briefly provide examples of the suggested methods and their applicability in the Ethiopian context?

Authors’ response: Thanks for this query. Now we have provided suggested methods and their applicability in Ethiopia as per your suggestion to make the discussion more clear. 

Comment 18: Compare Line 86 and 755 defining MHSU. Authors should stick to one MHSU abbreviation definition and not interchange between maternal health service use and Maternal Health Service Utilization. 

Authors’ response: Dear reviewer thank you for your important comment. Now we have used the term “Maternal Health Service Utilization” consistently throughout the manuscript. 

Comment 19: Line 105 – Edit the statement to “universal inaccessibility of maternal health services”

Authors’ response: Thank you for your vital comment and for fixing my mistake. Now we have revised it based on your edition.

Comment 20: Line 229 – Delete the word maintain

Authors’ response: Thank you for your suggestion, and we have deleted the word. 

Comment 21: Line 320 – Delete the space after discussants

Authors’ response: Thank you for your suggestion, and we have deleted the space.

Comment 22: Line 587 – Remove the apostrophe and letter ‘s’ from the word women’s

Authors’ response: Thank you for your suggestion, and we have removed the apostrophe and letter‘s’.

Reviewer 2 

Comment 1: Congratulations to the authors for this piece of work. I have the following comments that the authors should consider.

Authors’ response: Thank you for your kind remark. 

Comment 2: In the introduction section, provide some details about MHS being free. This is mentioned in the results but not in the background. This will help the reader with understanding the context better. 

Authors’ response: Thank you for this comment. Now we have provided some details about MHS being free in the introduction section to make the introduction more comprehensive and logical for the reader, as per your suggestion.

Comment 3: Sample size determination: There is repetition in the sample size determination for IDIs. This is mentioned in lines 168 - 172 and again in lines 186 - 191. Secondly, in the first instance the authors say they settled on 16 participants but this number changes to 15 in the second instance. Review this and align.

Authors’ response: Thank you for this comment. However, the sample size determination is not repeated. In the first section, we provided details of sample size determination for in-depth interviews, and in the in the second part, we discussed sample size determination for key informant interviews. The 16 study participants were fixed for in-depth interviews, while the 15 study participants were fixed for key informant interviews, so that there is no inconsistency. We have revised the sample size determination section based on your suggestion to make it clearer for the reader.

Key informant interviews (KIIs) and in-depth interviews (IDIs) have different sample size considerations. 

References 

1. 19.4 Phenomenology – Doctoral Research Methods in Social Work (pressbooks.pub) 

2. PNABS541.pdf (usaid.gov) 

3. s10508-012-0016-6.pdf (springer.com) 

4. Sample Size Policy for Qualitative Studies Using In-Depth Interviews | Archives of Sexual Behavior (springer.com) 

Comment 4: Provide more clarity on the difference between IDIs and KIIs and who they were conducted with.

Authors’ response: Thanks for this suggestion. As indicated in the above comment, we have provided more details on the difference between IDIs and KIIs as per your suggestion. The key differences between IDIs and KIIs are based on four parameters: expertise, goal, time, and bias. KIIs are experts. Thus, researchers only conduct KIIs when they can find a person with unique expertise on a topic. IDIs can be conducted with anyone. IDIs are used to learn more about a previously discussed problem. KIIs, on the other hand, are intended to examine a topic before delving into the details. It's often the initial step: figure out what has to be investigated! KIIs often require less time to conduct than traditional IDIs. This is because there are often fewer "experts" to interview than the general population (a sample of participants may be chosen for IDIs). Frequently, KIIs do not accurately represent a population. They are knowledgeable, but it may not translate well into the actual world. This is something that researchers should keep in mind while selecting KIIs.

Comment 5: In line 249 the authors mention that probing questions were asked to capture experiences and perceptions. This can be understood as meaning that only the probing questions were aimed at bringing these out while the study is about experiences and perceptions and as such all questions would be aimed at this. The authors can rephrase this. Probing questions are used to bring out more details/information in instances where more clarity or explanations are needed.

Authors’ response: Thank you for your vital comment and for fixing my mistake. Now we have revised it based on yo

---

## [Editor Report · Decision Letter 1]

2 Sep 2024

PONE-D-24-12696R1Perceptions, barriers, and facilitators of maternal health service utilization in southern Ethiopia: A qualitative exploration of community members’ and health care providers’ viewsPLOS ONE

Dear Dr. Yoseph ,

Thank you for submitting your manuscript to PLOS ONE. After careful consideration, we feel that it has merit but does not fully meet PLOS ONE’s publication criteria as it currently stands. Therefore, we invite you to submit a revised version of the manuscript that addresses the points raised during the review process.

We look forward to receiving your revised manuscript.

Kind regards,

Jackline Oluoch-Aridi, Ph.D.

Academic Editor

PLOS ONE

Journal Requirements:

Additional Editor Comments:

Dear authors,

Thank you for providing your revised manuscript after the two reviewers assessed your work. However there are a couple of improvements that still need to be done to make the article more readeable. I provide them below;

Abstract

The abstract section line 22 on methods should avoid showing the dates and can be described more succintly by saying the study was conducted in the month of November. Kindly change that

In Line 31 MAXQDA has been written as MAXQAD that too should be changed.

In line 86 you use unconventional abbreviations such as MHSU for Maternal Health Service Utilization please avoid this and use the correct verbs to describe utilization. You can easily describe utilization without the initials.

Lines 346-349 should be contained in a table of characterisitics of the respondents. This will make it more readable.

---

## [Author Response · Author response to Decision Letter 1]

12 Sep 2024

Point-by-point responses to editor’s comments

Comment 1: The abstract section line 22 on methods should avoid showing the dates and can be described more succinctly by saying the study was conducted in the month of November. Kindly change that

Authors’ response: Thank you a lot for this valuable comment. We have accepted the comment and made the required revision. 

Comment 2: In Line 31 MAXQDA has been written as MAXQAD that too should be changed.

Authors’ response: Thank you for your vital comment and for fixing our mistake. Now we have revised it based on your edition in the abstract and data analysis section of methods. 

Comment 3: In line 86 you use unconventional abbreviations such as MHSU for Maternal Health Service Utilization please avoid this and use the correct verbs to describe utilization. You can easily describe utilization without the initials.

Authors’ response: Dear Editor, thank you for your important comment. Now we have used the term “Maternal Health Service Use” consistently throughout the manuscript.

Comment 4: Lines 346-349 should be contained in a table of characteristics of the respondents. This will make it more readable.

Authors’ response: Dear Editor, thank you for your critical comment. Now we have provided the table that indicates characteristics of the study respondents, as per your suggestion to make the results more visible for readers.

Journal requirements

Requirement: Please review your reference list to ensure that it is complete and correct. If you have cited papers that have been retracted, please include the rationale for doing so in the manuscript text, or remove these references and replace them with relevant current references. Any changes to the reference list should be mentioned in the rebuttal letter that accompanies your revised manuscript. If you need to cite a retracted article, indicate the article’s retracted status in the References list and also include a citation and full reference for the retraction notice.

Authors’ response: All required revisions have been done to ensure that all references utilized in this manuscript are complete and correct, and the manuscript is updated accordingly. Now we are sure that all references are complete and correct and that no retracted articles were included.

---

## [Editor Report · Decision Letter 2]

8 Oct 2024

Perceptions, barriers, and facilitators of maternal health service utilization in southern Ethiopia: A qualitative exploration of community members’ and health care providers’ views

PONE-D-24-12696R2

Dear Dr. Yoseph ,

We’re pleased to inform you that your manuscript has been judged scientifically suitable for publication and will be formally accepted for publication once it meets all outstanding technical requirements.

Kind regards,

Jackline Oluoch-Aridi, Ph.D.

Academic Editor

PLOS ONE
---

## [Editor Report · Acceptance letter]

10 Oct 2024

PONE-D-24-12696R2 

PLOS ONE

Dear Dr. Yoseph , 

I'm pleased to inform you that your manuscript has been deemed suitable for publication in PLOS ONE. Congratulations! Your manuscript is now being handed over to our production team.

Kind regards, 

on behalf of

Dr. Jackline Oluoch-Aridi 

Academic Editor

PLOS ONE